

# Adsorption of phenol over bio-based silica/calcium carbonate (CS-SiO₂/CaCO₃) nanocomposite synthesized from waste eggshells and rice husks

Ibrahim Birma Bwatanglang, Samuel T. Magili and Iliya Kaigamma

Department of Pure and Applied Chemistry, Faculty of Science, Adamawa State University, Mubi, Mubi, Adamawa, Nigeria

## ABSTRACT

A bio-based Silica/Calcium Carbonate (CS–SiO₂/CaCO₃) nanocomposite was synthesized in this study using waste eggshells (ES) and rice husks (RH). The adsorbents (ESCaCO₃, RHSiO₂ and, CS-SiO₂/CaCO₃) characterized using XRD show crystallinity associated with the calcite and quartz phase. The FTIR of ESCaCO₃ shows the $CO_3^{-2}$ group of CaCO₃, while the spectra of RHSiO2 majorly show the siloxane bonds (Si–O–Si) in addition to the asymmetric and symmetric bending mode of SiO₂. The spectra for Chitosan (CS) show peaks corresponding to the C=O vibration mode of amides, C–N stretching, and C–O stretching. The CS–SiO₂/CaCO₃ nanocomposite shows the spectra pattern associated with ESCaCO₃ and RHSiO₂. The FESEM micrograph shows a near monodispersed and spherical CS–SiO₂/CaCO₃ nanocomposite morphology, with an average size distribution of 32.15 ± 6.20 nm. The corresponding EDX showed the representative peaks for Ca, C, Si, and O. The highest removal efficiency of phenol over the adsorbents was observed over CS–SiO₂/CaCO₃ nanocomposite compared to other adsorbents. Adsorbing 84–89% of phenol in 60–90 min at a pH of 5.4, and a dose of 0.15 g in 20 ml of 25 mg/L phenol concentration. The result of the kinetic model shows the adsorption processes to be best described by pseudo-second-order. The highest correlation coefficient ($R^2$) of 0.99 was observed in CS-SiO₂/CaCO₃ nanocomposite, followed by RHSiO₂ and ESCaCO₃. The result shows the equilibrium data for all the adsorbents fitting well to the Langmuir isotherm model, and follow the trend CS-SiO₂/CaCO₃ > ESCaCO₃ > RHSiO₂. The Langmuir equation and Freundlich model in this study show a higher correlation coefficient ($R^2$ = 0.9912 and 0.9905) for phenol adsorption onto the CS–SiO₂/CaCO₃ nanocomposite with a maximum adsorption capacity ($q_m$) of 14.06 mg/g compared to RHSiO₂ (10.64 mg/g) and ESCaCO₃ (10.33 mg/g). The results suggest good monolayer coverage on the adsorbent's surface (Langmuir) and heterogeneous surfaces with available binding sites (Freundlich).

# INTRODUCTION

The synthesis of adsorbents with adequate adsorption capacity and functionality underscores the need to efficiently remove emerging pollutants in water sources for human

Corresponding author
Ibrahim Birma Bwatanglang,
ibbbirma@gmail.com

consumption (*Daer et al., 2015*; *Pandey, Shukla & Singh, 2017*; *Husein, Hassanien & Al-Hakkani, 2019*). These emerging pollutants, if not removed in water sources for human consumption, could, in the long run, interfere and mimic physiological processes that regulate the metabolic functions of enzymes, hormones, and several biochemical indices (*Ali, AL-Othman & Alwarthan, 2016*). One of the emerging chemicals classified as a priority pollutant is phenol (*United States Environmental Protection Agency (USEPA), 1985*), with a recommended permissible limit of 1.0 μg/L allowed in drinking water (*Khare & Kumar, 2012*). Phenol is a common raw material for a range of pharmaceutical products, pesticides, fertilizers, disinfectants, and a host of household preservatives and detergents (*Ge et al., 2019*). The fingerprint of phenolic compounds is manifested in humans through several exposure pathways, exerting their toxicity by inducing protoplasmic poison, denaturing proteins, and creating an acid-base imbalance amongst many (*Agency for Toxic Substances & Disease Registry, 2014*).

In response to these health-related complications, scientist reported several techniques such as chemical oxidation, electrocoagulation, solvent extraction, membrane separation, and adsorption methods to be efficient in removing phenol from polluted water (*Burghoff & De Haan, 2009*; *Amin, Akhtar & Rai, 2010*; *El-Ashtoukhy et al., 2013*). Besides the adsorption methods, the other techniques highlighted above are characterized to be cost-driven, energy-intensive techniques, and generate toxic sludge in the process (*El-Ashtoukhy et al., 2013*). The adsorption processes are practically in vogue and utilized by many researchers to remove pollutants in water. Chief among the adsorption materials is activated carbon, which later becomes less economically viable as an adsorbent due to its costly regenerating processes, and relatively expensive starting material (*Khare & Kumar, 2012*; *Jin & Zhu, 2014*; *Ge et al., 2019*). Furthermore, adsorbent prepared from nature-based carbon materials (clay, peat, wood, sawdust, fly ash, coal reject, bagasse, and pine cones) has to undergo surface modifications to be efficient (*Djebbar et al., 2012*; *Chen et al., 2017*).

Therefore, this study attempts to create a biobased organic-inorganic framework to synthesize adsorbent materials using available nature-based materials that are sustainable (*Habte et al., 2019*). The study integrates the high surface area and facile/tunable adsorption-desorption characteristics of silica (*Chen et al., 2017*), the compatibility, better adhesion, and thermal stability of $CaCO_3$ (*Hassan, Rangari & Jeelani, 2014*), the natural cationic potential, and rich hydroxyl (–OH) and amino (–NH$_2$) groups of chitosan (*Dehaghi et al., 2014*) in the synthesis of the biosorbent material. Studies show that some plant species' tissues contain high biogenic silica deposits, mostly in its hydrated silica ($SiO_2.nH_2O$) form (*Ghorbani, Sanati & Maleki, 2015*). Similarly, eggshells, among other sources (cockle shells), are among the most readily available calcium carbonate sources in nature (*Hassan, Rangari & Jeelani, 2014*). Another material of choice used in the synthesis is chitosan, a natural cationic cellulose biopolymer derived from Crustaceans (*Bwatanglang et al., 2016*). These materials are of great environmental importance and are widely covered in green chemistry in the synthesis of adsorbent materials for the removal of phenol (*Giraldo & Moreno-Piraján, 2014*; *Chraibi et al., 2016*; *Sarker & Fakhruddin, 2017*; *Asgharnia et al., 2019*; *Mandal, Mukhopadhyay & Das, 2019*).

Other studies also reported using the same raw materials to synthesize biosilica and calcium carbonate $SiO_2/CaCO_3$ nanocomposites. *Morsy, El-Sheikh & Barhoum (2019)* used semi-burned rice straw ash to synthesize modified papermaking fillers nanocomposites by sol-gel technique. Similarly, *Hassan, Rangari & Jeelani (2014)* used waste eggshell to synthesize $CaCO_3$ nanoparticles reinforcement fillers prepared using a combination of mechanochemical and ultrasonic irradiation techniques. In the light of the above studies, this present work utilizes RH to synthesis $SiO_2$ and waste eggshells to derive the $CaCO_3$. The $SiO_2/CaCO_3$ was synthesis by stabilizing the particles in chitosan to form the $CS$-$SiO_2/CaCO_3$ nanocomposite not as fillers but instead as an adsorbent material for the removal of phenol. Therefore, this study's primary focus is to utilize these nature-based waste materials to design, synthesize, and formulate the adsorbing materials consisting of the organic phase (chitosan) and an inorganic phase (calcium carbonate and silica) for the removal of phenol in water.

## MATERIALS AND METHODS

### Chemical and reagents

The waste eggshells (ES) and rice husks (RH) used in this work were obtained from local fast-food restaurants in Mubi, Adamawa state Nigeria. The Low-molecular-weight Chitosan (CS) (75–85% degree of deacetylation) was purchased from Sigma-Aldrich (St Louis, MO, USA). Ethanol ($C_2H_5OH$, 99.7%), hydrochloric acid (HCl, 37%), sulfuric acid ($H_2SO_4$, 98.5%), hydrogen peroxide ($H_2O_2$), sodium hydroxide (NaOH, 97%), acetone ($C_3H_6O$, 99%), acetic acid ($CH_3COOH$, 99.85%) and phenol ($C_6H_6O$, 99%) were purchased from BDH Chemical Ltd England.

### Preparation and synthesis of the adsorbents

The preparation of the adsorbent follows sequential steps. Briefly, a thoroughly washed RH was subjected to acid pretreatment using HCl: $H_2SO_4$ (10:30 wt%) and further treatment with $H_2O_2$ (30% v/v) at 70 °C for 60 min (*Lu & Hsieh, 2012*; *Thuc & Thuc, 2013*). The formed slurry was rinsed with distilled water, oven-dried at 600 °C for 4 h. The sieved RH is then dispersed in an aqueous 0.5M NaOH solution under vigorous stirring for 1 h to form the RH-derived sodium silicate ($RH$-$Na_2SiO_3$). The RH-silicate ($RHSiO_2$) is obtained by subjecting $RH$-$Na_2SiO_3$, in 12 wt% $H_2SO_4$ under vigorous stirring for 15 min. The formed $RHSiO_2$ is washed in water and aged in ethanol at 60 °C for 60 min. The final material is further rinsed in water and dried under ambient temperature (*Lu & Hsieh, 2012*; *Thuc & Thuc, 2013*; *Ghorbani, Sanati & Maleki, 2015*). The proteins in the Chicken eggshells are deactivated/denatured by boiling after subjecting the same to washing using acetone/alcohol (1:1 ratio), dried, and made into powder. The powdered samples were further dispersed in distilled water under sonication, collected and heated in an oven at 600 °C for 3 h, and finally sieved using a 40–75 μm to form the ES-calcium carbonate ($ESCaCO_3$) particle (*Minakshi et al., 2019*).

An aqueous solution of $ESCaCO_3$ (5 wt%) was prepared under vigorous stirring at 3,000 rpm for 30 min. To the formed precursor, 1 g of $RHSiO_2$ in 20 ml of distilled water was added gradually under heating at 80 °C and stirred for 60 min to form the composite

mixture of RHSiO$_2$/ESCaCO$_3$ particles (*Morsy, El-Sheikh & Barhoum, 2019*).
The Bio-based CS-SiO$_2$/CaCO$_3$ nanocomposites were prepared by the dropwise addition of aqueous RHSiO$_2$/ESCaCO$_3$ (1 g/20 mL) solution into chitosan (CS) suspensions (0.1 g in 10 mL of 1% acetic acid) under stirring for 30 min in other to stabilize the particles. The formed precursor was incubated at 60 °C for 3 h, collected by centrifugation, and further allowed to age in 2.0M NaOH solution for 2 h. Then wash using distilled water and filtered to obtain the CS-SiO$_2$/CaCO$_3$ nanocomposites.

## Instrumental analysis

In this study, different instruments were used at various stages of the analysis to characterize the prepared samples. The structure and phase identification of the prepared RHSiO$_2$, ESCaCO$_3$, and the CS-SiO$_2$/CaCO$_3$ nanocomposites were established using an X-ray diffraction (XRD) instrument (Malvern PAAnalytical Empyrean, Nederland) equipped with Cu anode (λCuKα radiation source) X-ray tube. The identification of the functional groups was determined using Fourier Transform Infrared (FTIR) spectroscopy using PerkinElmer (Waltham, MA, USA, N3895) with a range covering 4,000–500 cm$^{-1}$. The elemental composition and morphology were studied using Field Emission Scanning Electron Microscopy and Energy Dispersive X-ray (FESEM/EDX) on JEOL JSM-7600F (JEOL, Tokyo, Japan).

## Adsorption experiment

The adsorbents were brought into contact with aqueous phenol with a concentration ranging from 10 to 30 mg/L and agitated at a speed of 120 rpm at room temperature. The percentage of phenol in each of the aliquot samples taken were determined after 30, 45, 60, 70, 80, and 90 min by measuring the difference in absorbance using UV–Vis Spectrophotometer (PerkinElmer Lambda 35 spectrometer). The study was conducted using changes in pH (4, 5.4, 6, 7.4, and 8) and adsorbent dose (0.05, 0.1, 0.15, 0.2, and 0.25 g). The adsorption kinetics were determined using the Pseudo-first-order and Pseudo-second-order kinetic model. The Pseudo-first-order kinetic model was determined using Eq. (1):

$$Log\ (qe - qt) = Logqe - \frac{K_1 t}{2.303} \tag{1}$$

where $K_1$ (mg/g min) is the rate constant, $qe$ (mg/g) is the adsorption capacity at equilibrium, and $qt$ (mg/g) is the adsorption capacity at time $t$. The slope of the *Log (qe − qt)* vs $t$ was used for the determination of the equilibrium rate constant $K_1$. The Pseudo-second order kinetic was determined using Eq. (2):

$$\frac{t}{qt} = \frac{1}{K_2 qe^2} + \frac{1}{qt}\ t \tag{2}$$

The rate constant ($K_2$) and $qe$ are calculated from the intercept and slope of the plot of $t/qt$ vs $t$ (*Husein, Hassanien & Al-Hakkani, 2019*). The percentage removal

efficiency of phenol and the amount adsorbed over the adsorbent were determined using Eq. (3):

$$R\% = \left[\frac{Ci - Ce}{Ci}\right] \times 100 \tag{3}$$

where $Ci$ and $Ce$ are the initial and the equilibrium concentrations of phenol in mg/L. The uptake capacity (mg/g) of the sorbent for each concentration of phenol at equilibrium were determined using Eq. (4):

$$qe\ (mg/g) = [Ci - Ce)/M] \times V \tag{4}$$

where $V$ is the volume of the solution in $L$, while $M$ is the mass of the biosorbent (g). The adsorption behavior was analyzed using The Langmuir (Langmuir, 1916) and Freundlich (Freundlich, 1906) isotherms models. The following relation represents the linear form of the Langmuir isotherm model:

$$\frac{Ce}{qe} = \frac{1}{qmKe} + \frac{Ce}{qm} \tag{5}$$

where $qe$ is the amount adsorbed at equilibrium (mg/g), $Ce$ is the equilibrium concentration of the adsorbate (mg/L), and $qm$ (mg/g) and $Ke$ (L/mg) are the Langmuir constants related to the maximum adsorption capacity and the energy of adsorption, respectively. These constants are evaluated from the intercept and slope of the linear plot experimental data of $Ce/qe$ (g/mg) vs $Ce$ (L/mg). The following relation gives the linear form of the Freundlich isotherm model:

$$Logqe = LogK_f + \frac{I}{n}LogCe \tag{6}$$

where $K_f$ and $1/n$ are the Freundlich constants related to adsorption capacity and adsorption intensity of the adsorbent. The values of $K_f$ and $1/n$ were derived from the intercept and slope of the linear plot of $Logqe$ vs $LogCe$. The dimensionless constant (separation factor, $R_L$) taking from the Langmuir isotherm was estimated using Eq. (7):

$$R_L = \frac{1}{[K_L Ci + 1]} \tag{7}$$

where $Ci$ is the initial concentration and $K_L$ the concentration of Langmuir. If $R_L = 0$, the adsorption is irreversible, is favorable when $0 < R_L < 1$, linear when $R_L = 1$ and unfavorable when $R_L > 1$ (Chraibi et al., 2016).

# RESULTS AND DISCUSSION

## Characterization of adsorbent

### FTIR analysis

The fingerprints or functional groups involved in the various stages of the synthesis identified by FTIR are shown in Fig. 1. In the figure, the broadband in the spectra of

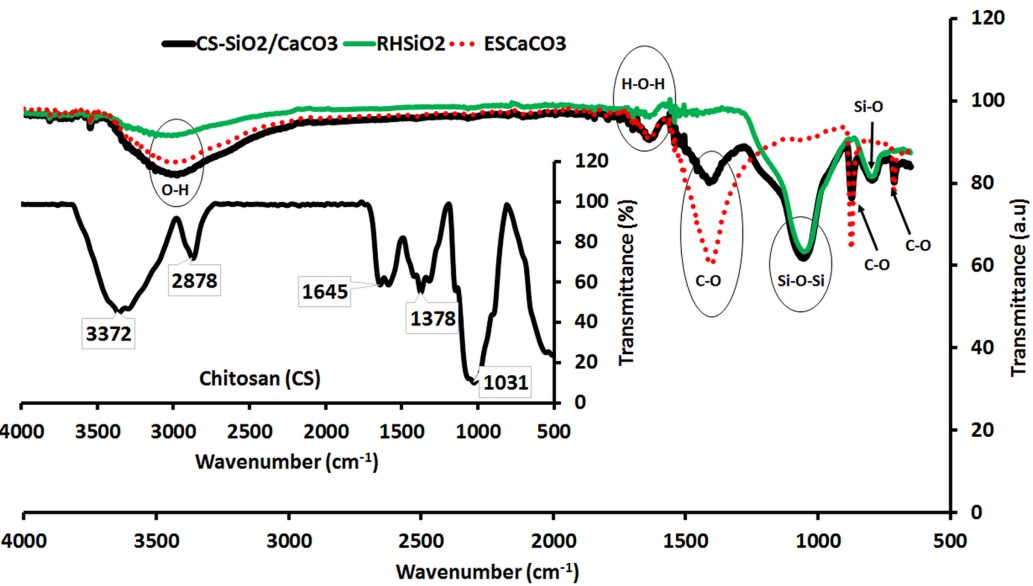

**Figure 1** The FTIR spectra of ESCaCO₃, RHSiO₂, CS–SiO₂/CaCO₃ nanocomposite and CS (Insert).

ESCaCO$_3$ centered around 3,104–3,630 cm$^{-1}$ is an O–H bond from water, while the peak at 1,651 cm$^{-1}$ is the adsorption band of H–O–H bending vibrations from adsorbed water molecules (*Ihli et al., 2014*; *Habte et al., 2019*). The three peaks at 1,413 cm$^{-1}$, 873 cm$^{-1}$, and 712 cm$^{-1}$ are absorption peaks due to the CO$_3^{-2}$ group in CaCO$_3$ (*Nyquist & Kagel, 2012*; *Morsy, El-Sheikh & Barhoum, 2019*; *Du & Amstad, 2020*). The peaks at 3,285–3,629 cm$^{-1}$, 1,651 cm$^{-1}$, 1,056 cm$^{-1}$, and 796 cm$^{-1}$ in the spectra of RHSiO2 are from the silanol OH groups of water, the H–O–H bending vibration, siloxane asymmetric bending vibrations bands (Si–O–Si), and the symmetric bending mode of silanol (Si–O) (*Kamath & Proctor, 1998*; *Ghorbani, Sanati & Maleki, 2015*). Chitosan's spectra show stretching vibration of OH group around 3,372 cm$^{-1}$ and a –CH$_2$ symmetric vibration at 2,878 cm$^{-1}$. The peaks at 1,645 cm$^{-1}$, 1,378 cm$^{-1}$, and the prominent peak at 1,031 cm$^{-1}$ are C=O vibration mode of amides, C–N stretching, and C–O stretching vibrations, respectively (*Saifuddin & Dinara, 2012*; *Dehaghi et al., 2014*).

The broadband around 3,312–3,618 cm$^{-1}$ and the peak centered at 1,643 cm$^{-1}$ are fingerprints from O–H bond and H–O–H bending vibrations of ESCaCO$_3$, and RHSiO$_2$, in the CS-SiO$_2$/CaCO$_3$ nanocomposite. The absorption peaks centered at 1,412 cm$^{-1}$ and 873 cm$^{-1}$ are from the CO$_3^{-2}$ group in ESCaCO$_3$, while the peaks at 1,065 cm$^{-1}$, 795 cm$^{-1}$, and 713 cm$^{-1}$ are from the Si–OH stretching, Si–O–Si asymmetric, and Si–O–Si symmetric stretching vibrations from the RHSiO$_2$ respectively (*Nyquist & Kagel, 2012*; *Morsy, El-Sheikh & Barhoum, 2019*). The Si–O–Si asymmetric vibrational mode from the spectra of RHSiO$_2$ centered at 1,056 cm$^{-1}$ slightly shifted to 1,065 cm$^{-1}$ in the nanocomposites, suggesting a possible interaction with the C–O stretching vibrations of CS (1,031 cm$^{-1}$).

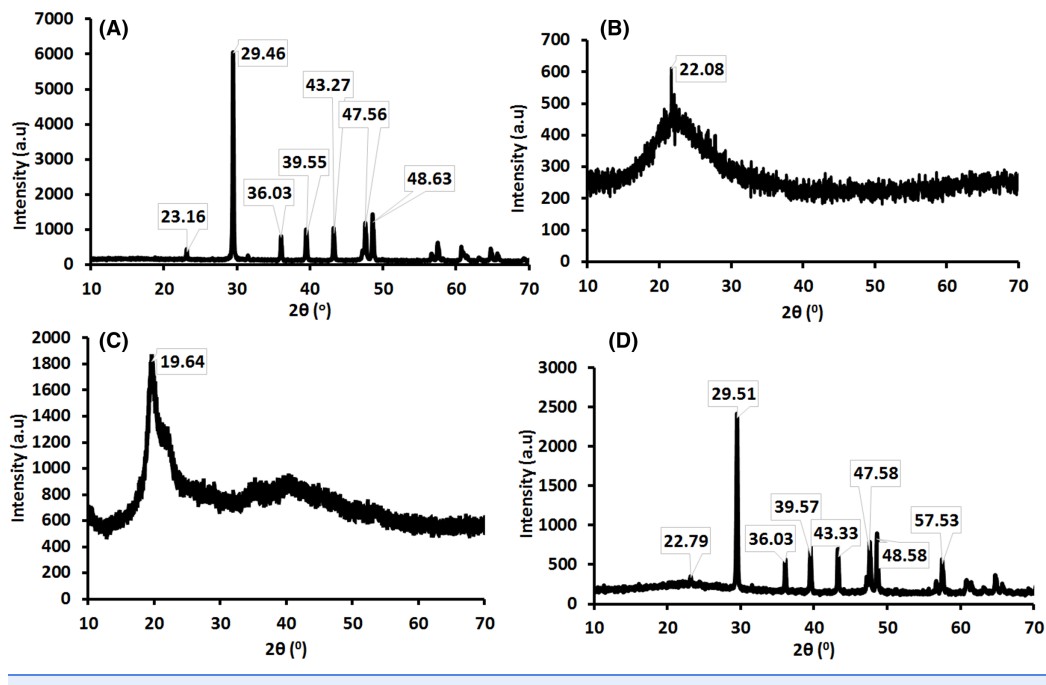

**Figure 2 X-ray diffractogram of (A) ESCaCO₃ (B) RHSiO₂ (C) CS and (D) CS–SiO₂/CaCO₃ nanocomposite.**

### XRD analysis

Figure 2 represents the XRD patterns of (a) ESCaCO$_3$, (b) RHSiO$_2$, (c) CS and, (d) CS–SiO$_2$/CaCO$_3$ nanocomposite. Figure 2A shows the XRD patterns for ESCaCO$_3$ having all the diffraction peaks corresponding to the standard pattern for calcite (JCPDS card no. 005-0586), indicating 81% CaCO$_3$ (*Hassan, Rangari & Jeelani, 2014*; *Minakshi et al., 2019*). The maximum peaks at 2θ = 29.46° reflect the calcite phase of CaCO$_3$. This result agrees with the findings reported by *Chraibi et al. (2016)*, showing the peak at 2θ = 29.48° for CaCO$_3$ derived from the eggshell. Figure 2B shows the XRD patterns of silica prepared from rice husk (RHSiO$_2$). The peak corresponding to the semi-crystalline phase of Si appeared at a broad peak centered at 2θ = 22.08°. The analysis of the particles indicates the presence of silica (96%). The result agrees with the study by *Morsy, El-Sheikh & Barhoum (2019)* showing 82% silica and a broad peak centered at 2θ = 22.5° corresponding to silica nanoparticles' semi-crystalline phase. A strong peak centered at ~20° was observed in the spectra of pure chitosan (Fig. 2C). This high degree of crystallinity was observed to transcend to a very weak peak that is not noticeable in the spectra of CS–SiO$_2$/CaCO$_3$ nanocomposite, arising from possible disarray in the chain alignment of chitosan by the diffraction peaks of calcite and the semi-crystalline phase of Si (*Dehaghi et al., 2014*). Furthermore, Fig. 2D shows the XRD patterns of the CS–SiO$_2$/CaCO$_3$ nanocomposites. The results show the diffraction peaks corresponding to the calcite phase of CaCO$_3$ at 2θ = 29.51° and a semi-crystalline phase related to silica at 2θ = 22.79°. The result further shows that CS–SiO$_2$/CaCO$_3$ nanocomposite predominantly consists of 9% silica and 52% CaCO$_3$.

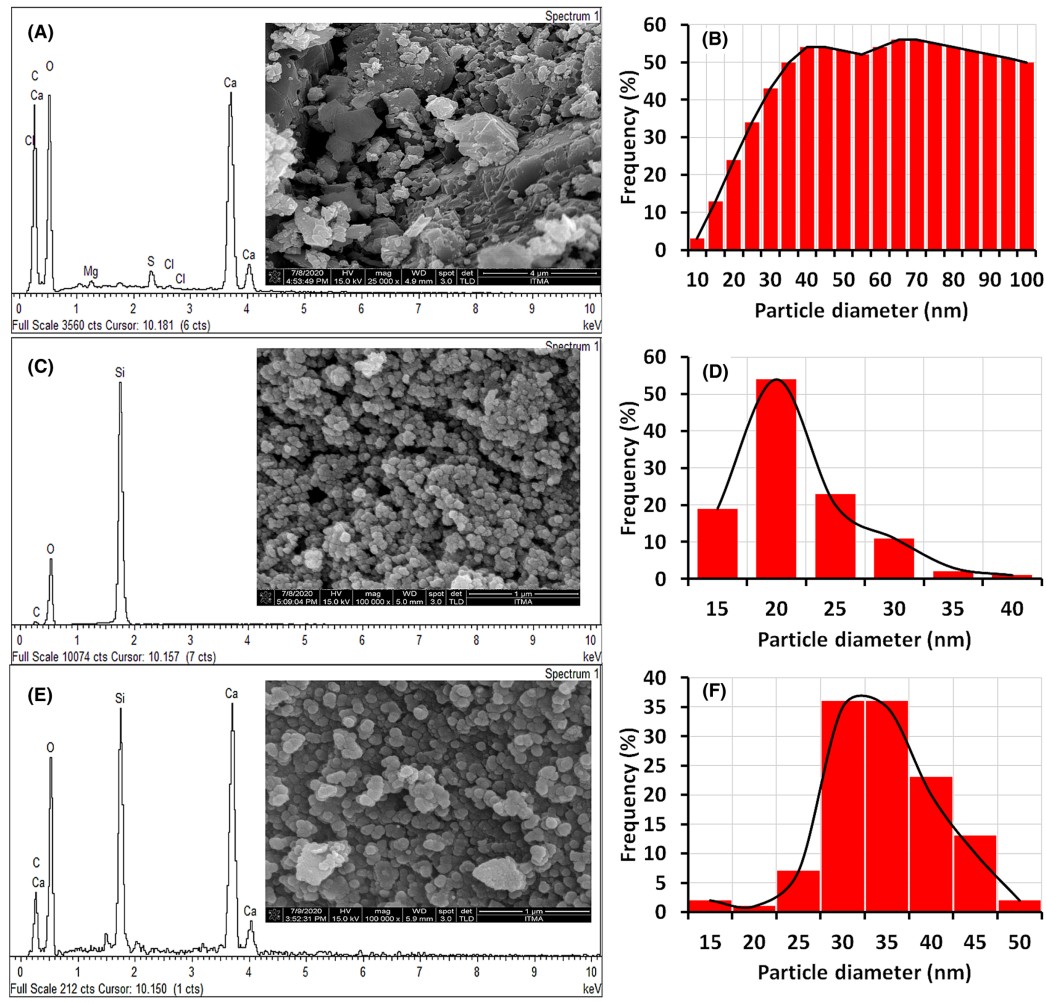

**Figure 3 FESEM macrograph and EDX spectra of (A) ESCaCO₃, (C) RHSiO₂, (E) CS–SiO₂/CaCO₃ nanocomposite and their corresponding particle size (B, D and F).**

### FESEM/EDX analysis

Figure 3 shows FESEM/EDX micrographs with the corresponding particle size distribution of ESCaCO₃, RHSiO₂, and CS-SiO₂/CaCO₃ nanocomposites. The micrograph of ESCaCO₃, as shown in Fig. 3A, reveals an irregular surface structure with a random assembly of aggregated grains that form a densely packed structure and size distribution up to 100 nm (Fig. 3B). Similar morphological features were reported by *Minakshi et al. (2019)* and *Ahmad et al. (2020)*, showing the structure of eggshells-CaCO₃ having an irregular surface structure with a size distribution ~200 nm and 89 nm, respectively. The morphology of the RHSiO₂, as shown in Fig. 3C, shows nearly spherical densely packed grains with an average particle size of 19.07 ± 4.63 nm (Fig. 3D). The images are similar to those reported by *Thuc & Thuc (2013)*, *Morsy, El-Sheikh & Barhoum (2019)*, and *Phoohinkong & Kitthawee (2014)*. The result in Fig. 3E shows that the CS-SiO₂/CaCO₃ nanocomposite have better and good monodispersed spherical morphology than those of

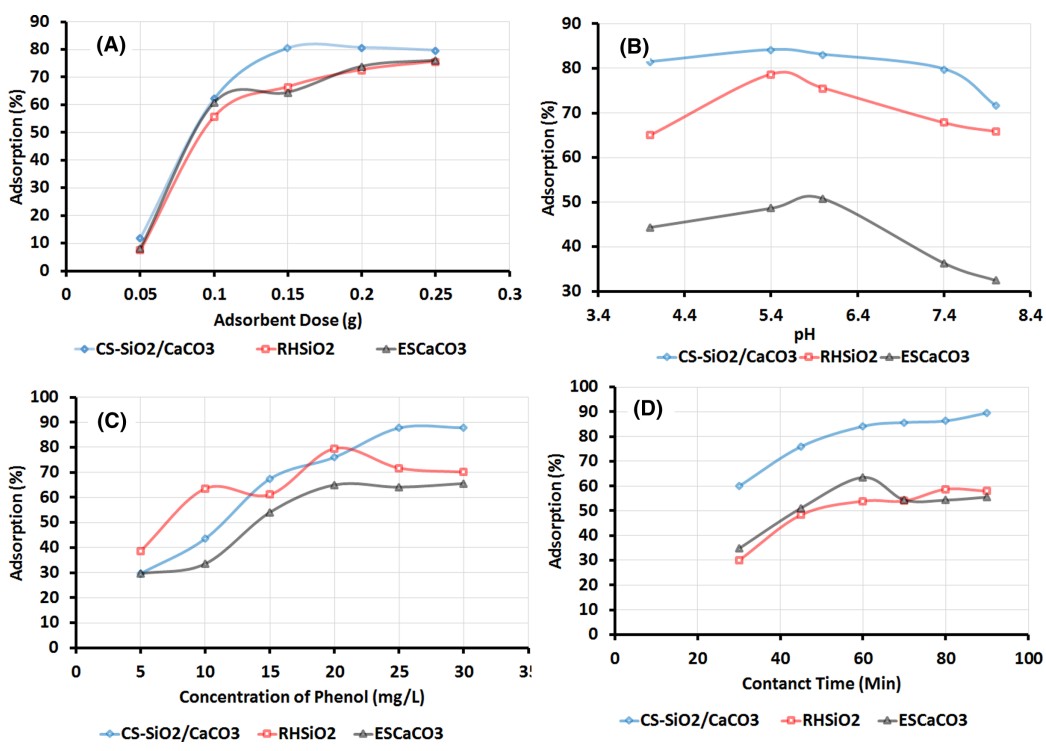

**Figure 4 Effect of varying (A) adsorbent dose, (B) pH, (C) phenol concentrations, (D) contact time on the adsorption of phenol over ESCaCO₃, RHSiO₂ and CS-SiO₂/CaCO₃ nanocomposite.**

the single EsCaCO₃ particles. The dispersion of the SiO₂/CaCO₃ into CS suspension increases the uniformity of the grain size of CS–SiO₂/CaCO₃, with an average size distribution of 32.15 ± 6.20 nm (Fig. 3F). The result falls near to the findings reported by *Morsy, El-Sheikh & Barhoum (2019)*, showing SiO₂/CaCO₃ nanocomposites to be nearly spherical with particle sizes ranging 70–110 nm. The corresponding EDX results, as presented in Figs. 3A, 3C, and 3E, showed representative peaks for Ca, Si, and O for all the samples. The EDX of ESCaCO₃ consists mainly of C (36%), O (52%), Ca (10%), while the spectra of RHSiO₂ show majorly the peaks of Si (30%) and O (69). The CS-SiO₂/CaCO₃ nanocomposite show similar peaks corresponding to the spectra of SiO2 and CaCO3, which indicate a successful formation of CS-SiO₂/CaCO₃ nanocomposite.

## Factors affecting adsorption behaviors

The Phenol removal efficiency of the adsorbent, as shown in Fig. 4A, shows the adsorption capacity increases in the order of CS-SiO₂/CaCO₃ > RHSiO₂ > ESCaCO₃ with an increase in the adsorbent dose. The synthesized CS-SiO₂/CaCO₃ nanocomposite due to additional binding sites from SiO₂, CaCO₃, and CS adsorbed more phenol than the RHSiO₂ and ESCaCO₃ particles. The study was conducted at a varying initial dosage of 0.05–0.25 g, pH 5.4, and initial phenol concentration of 25 mg/L at 120 rpm for 60 min. The highest removal efficiency of 80% was achieved at a dose of 0.15 g of CS-SiO₂/CaCO₃ nanocomposite and remain more or less the same up to 0.25 g. Suggesting that the

removal efficiency at the onset of the adsorption processes was higher due to the more available surface area, additional adsorption sites, and then remained the same due to saturating absorbent sites following an increase in the dose (*Mandal, Mukhopadhyay & Das, 2019*). The RHSiO$_2$ and ESCaCO$_3$ at varying initial dosage show an increase in phenol removal efficiency. The highest removal efficiency of phenol was observed at a dose of 0.25 g of RHSiO$_2$ (75%) and ESCaCO$_3$ (76%), respectively. An increase in phenol removal efficiency with an increase in adsorbent dosage was also observed and reported by *Sarker & Fakhruddin (2017)* and *Asgharnia et al. (2019)* using rice straw as adsorbent. Similar observations were reported using CaCO$_3$ derived from the eggshell. The study reported an optimum dosage of eggshell powder adsorbent at 4 g (*Kashi, 2017*).

Figure 4B described the adsorption efficiency of the adsorbent at varying pH environments. The study was carried out at varying pH 4, 5.4, 6, 7.4, and 8, with an initial phenol concentration of 25 mg/L and an adsorbent dose of 0.15 g at 120 rpm for 60 min. The plots show higher removal efficiency occurring at acidic pH range (4–6) and decreasing at pH of 7.4 and 8. At these pH ranges, the acidic environment substantially increases the electrostatic attraction with the phenolates' negative ions, increasing phenol adsorption in the process (*Khare & Kumar, 2012*; *Daraei et al., 2013*; *Ouallal et al., 2019*). The adsorption processes following the pH variation were observed to follow the trend CS–SiO$_2$/CaCO$_3$ > RHSiO$_2$ > ESCaCO$_3$. The highest percentage removal of 83% was observed in the final nanocomposite (CS–SiO$_2$/CaCO$_3$) at a pH of 5.4. The presence of available exchangeable ions (*Kashi, 2017*), multiple adsorption sites on the adsorbent at this pH range leads to the observed removal efficiency in the CS–SiO$_2$/CaCO$_3$ nanocomposite compared to the other adsorbents. *Asgharnia et al. (2019)*, while investigating the effect of pH on phenol removal efficiency, observed higher efficiency for rice husk activated carbon (RHAC) at pH 6 and rice husk carbon (RHC) at pH 5. The result further agrees with the findings reported by *Kashi (2017)* and *Daraei et al. (2013)* using powdered eggshell.

Varying the initial concentration of phenol from 5–30 mg/L, as shown in Fig. 4C, shows the removal efficiency increases as the concentration increases from 5–25 mg/L and remains the same at 30 mg/L. The CS-SiO$_2$/CaCO$_3$ nanocomposite gives the highest removal efficiency (87%) at a concentration range of 25 mg/L and 30 mg/L. The removal efficiency follows the trend of CS-SiO$_2$/CaCO$_3$ > RHSiO$_2$ > ESCaCO$_3$, at a fixed adsorbent dose of 0.15 g, pH 5.4 at 120 rpm for 60 min. An increase in concentration leads to a rise in phenol adsorption on the surface of the sorbent. But above 25 mg/L, the removal efficiency remains the same, which suggests a saturation of the sorption sites available for sorption at this concentration of the sorbate molecules (*Dehaghi et al., 2014*; *Sarker & Fakhruddin, 2017*).

Figure 4D shows the effect of contact time (30, 45, 60, 70, 80, and 90) on the percentage removal of phenol using 0.15 g of adsorbent in 20 ml of 25 mg/L phenol, pH 5.4 at 120 rpm. The result shows an increase in phenol removal efficiency after 60 min of contact with CS–SiO$_2$/CaCO$_3$ nanocomposite and RHSiO$_2$ particles. The CS-SiO$_2$/CaCO$_3$ nanocomposite shows removal efficiency of 84% after 60 min and 89% at 90 min of contact

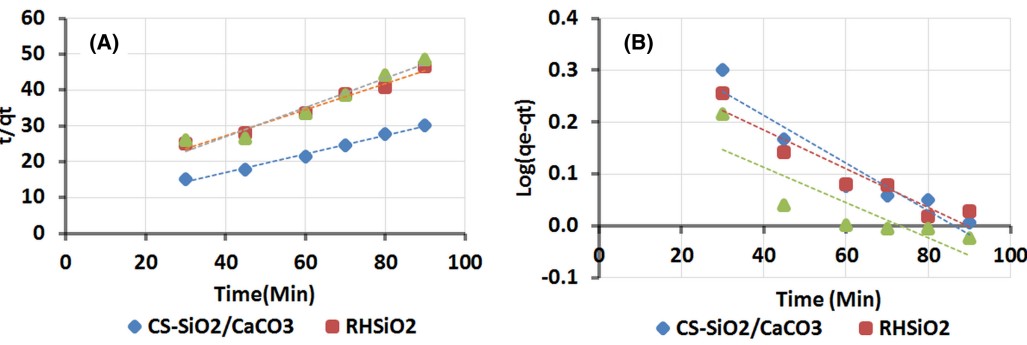

**Figure 5 Kinetic adsorption study of phenol (A) Pseudo-second order model (B) Pseudo-first-order model.** Adsorption condition: pH 5.4, phenol concentration of 25 mg/L, adsorbent dose of 0.15 g at 120 rpm.

**Table 1 Pseudo-first-order and Pseudo-second order constants for adsorption of phenol.**

| Adsorbents | Pseudo-first-order constants | | | Pseudo-second order constants | | |
|---|---|---|---|---|---|---|
| | $qm$ (mg/g) | $K_1$ (min-1) | $R^2$ | $qm$ (mg/g) | $K_2$ (g/mg min) | $R^2$ |
| $CS-SiO_2/CaCO_3$ | 2.49 | −1.1E−02 | 0.91 | 3.80 | 0.01 | 0.99 |
| $RHSiO_2$ | 2.20 | −9.0E−03 | 0.90 | 2.80 | 0.01 | 0.98 |
| $ESCaCO_3$ | 1.80 | −7.8E−03 | 0.72 | 2.26 | 0.02 | 0.95 |

time. The result agrees with findings reported by *Asgharnia et al. (2019)* and *Sarker & Fakhruddin (2017)* using RH as an adsorbent.

## Kinetic adsorption study

To further understand phenol adsorption behavior and mechanism onto $CS–SiO_2/CaCO_3$, $RHSiO_2$, and ESCaCO3, the study was subjected to kinetic analysis using pseudo-first-order and pseudo-second-order models. The models help to predict possible interaction mechanisms between the adsorbent and the adsorbate. The result is shown in Fig. 5 and further elaborated in Table 1. The plot from the graph of $t/qt$ vs $t$ fit well with pseudo-second-order with the highest correlation coefficient ($R^2$) of 0.99 and $qm$ values of 3.80 mg/g observed in $CS–SiO_2/CaCO_3$ nanocomposite (Fig. 5A). The result further shows phenol adsorption over $RHSiO_2$ ($R^2 = 0.98$) and $ESCaCO_3$ ($R^2 = 0.95$) fitting well with the pseudo-second-order. The outcome suggests that the adsorption process is chemisorption and the adsorbent particles are heterogeneous (*Mandal, Mukhopadhyay & Das, 2019*). On the other hand, the pseudo-first-order graph of $log (q_e−q_t)$ against $t$, as shown in Fig. 5B, poorly described the absorption processes of $ESCaCO_3$ ($R^2 = 0.72$), but a bit linear for $CS-SiO_2/CaCO_3$ nanocomposite ($R^2 = 0.91$) and $RHSiO_2$ ($R^2 = 0.90$) respectively. Pseudo-first-order suggests that the adsorption process is physisorption and the adsorbent particles are homogeneous (*Husein, Hassanien & Al-Hakkani, 2019*). This study's result agrees with the findings reported by *Mandal, Mukhopadhyay & Das (2019)*. Reporting pseudo-second-order supporting the adsorption processes of phenol using rice husk as adsorbent. Though pseudo-second-order best described the adsorption

processes, the $R^2$ derived from the pseudo-first-order model of CS–SiO$_2$/CaCO$_3$ and RHSiO$_2$, though poorly described the adsorption process further characterize the homogeneity or the heterogeneity of the adsorption sites (*Ouallal et al., 2019*). These observations also suggest some semblance of pseudo-first-order mechanism influencing the adsorption process of CS–SiO$_2$/CaCO$_3$ nanocomposite and RHSiO$_2$ particles. Which further aggress with isotherm result from this study, showing the adsorption processes to possess good monolayer coverage on the surface and heterogeneous surface with different binding sites for CS–SiO$_2$/CaCO$_3$ nanocomposite (*Song, Johnson & Elimelech, 1994*). The higher values of $K_2$, compared to $K_1$ (Table 1), further suggest a pseudo-first-order reaction's applicability (*Ali, AL-Othman & Alwarthan, 2016*).

## Adsorption isotherms study

As presented in Fig. 6, the result showed the equilibrium data for all the adsorbents at an initial concentration of 5–30 mg/L, demonstrating a better matching to the Langmuir isotherm model. Observed to follow the trend CS–SiO$_2$/CaCO$_3$ > ESCaCO$_3$ > RHSiO$_2$. *Chraibi et al. (2016)* and *Kashi (2017)*, in a separate study, show Langmuir isotherm describing the adsorption of phenol onto a calcined eggshell. *Giraldo & Moreno-Piraján (2014)* observed similar findings over activated carbons derived from eggshell particles. *Asgharnia et al. (2019)* also reported a better matching of phenol adsorption with the Langmuir isotherm model using RHS and RHAC as adsorbents.

Except for CS–SiO$_2$/CaCO$_3$ nanocomposite (Fig. 6A), the adsorption data for ESCaCO$_3$ (Fig. 6B) and RHSiO$_2$ (Fig. 6C) base on their $R^2$ values, fitted poorly to the Freundlich isotherm model. However, the magnitude of their $K_F$ (4.60 and 3.3) showed the tendency of phenol adsorption over the adsorbents (*Khare & Kumar, 2012*). The Langmuir equation and Freundlich model in this study show a higher correlation coefficient ($R^2 = 0.9912$ and 0.9905) for phenol adsorption onto CS–SiO$_2$/CaCO$_3$ nanocomposite (Fig. 6A). Suggesting good monolayer coverage on the surface of the adsorbent (Langmuir) and equally possess heterogeneous surfaces with available binding sites (Freundlich) (*Kermani et al., 2012*). The $K_F$ (3.4) and $1/n$ (2.5) values calculated from the Freundlich isotherm for CS–SiO$_2$/CaCO$_3$ nanocomposite further suggest that the sorption process is complimentary and physicochemical (*Song, Johnson & Elimelech, 1994*). The magnitude of $1/n$ indicates that phenol was favorably adsorbed by CS–SiO$_2$/CaCO$_3$ nanocomposite compared to the other adsorbent (*Ouallal et al., 2019*). From the data, the magnitude of $qm$, which defines the amount of phenol per unit weight of sorbent, was higher for CS–SiO$_2$/CaCO$_3$ nanocomposite in comparison to RHSiO$_2$ and ESCaCO$_3$ particles. Similar results were also reported by *Kermani et al. (2012)* using rice husk ash as an adsorbent. In the study, both the Langmuir and Freundlich isotherm model adequately described the adsorption process. *Djebbar et al. (2012)* reported similar findings for phenol's adsorption using natural clay.

The $R_L$ estimation at various $C_i$ further shows the adsorption to be more favored by the Langmuir isotherm model, having values of <1 for all the adsorbents. The results indicate favorable adsorption in all cases. Similar observations have been reported for phenol's sorption on rice husk and chitin (*Milhome et al., 2009*; *Chraibi et al., 2016*).

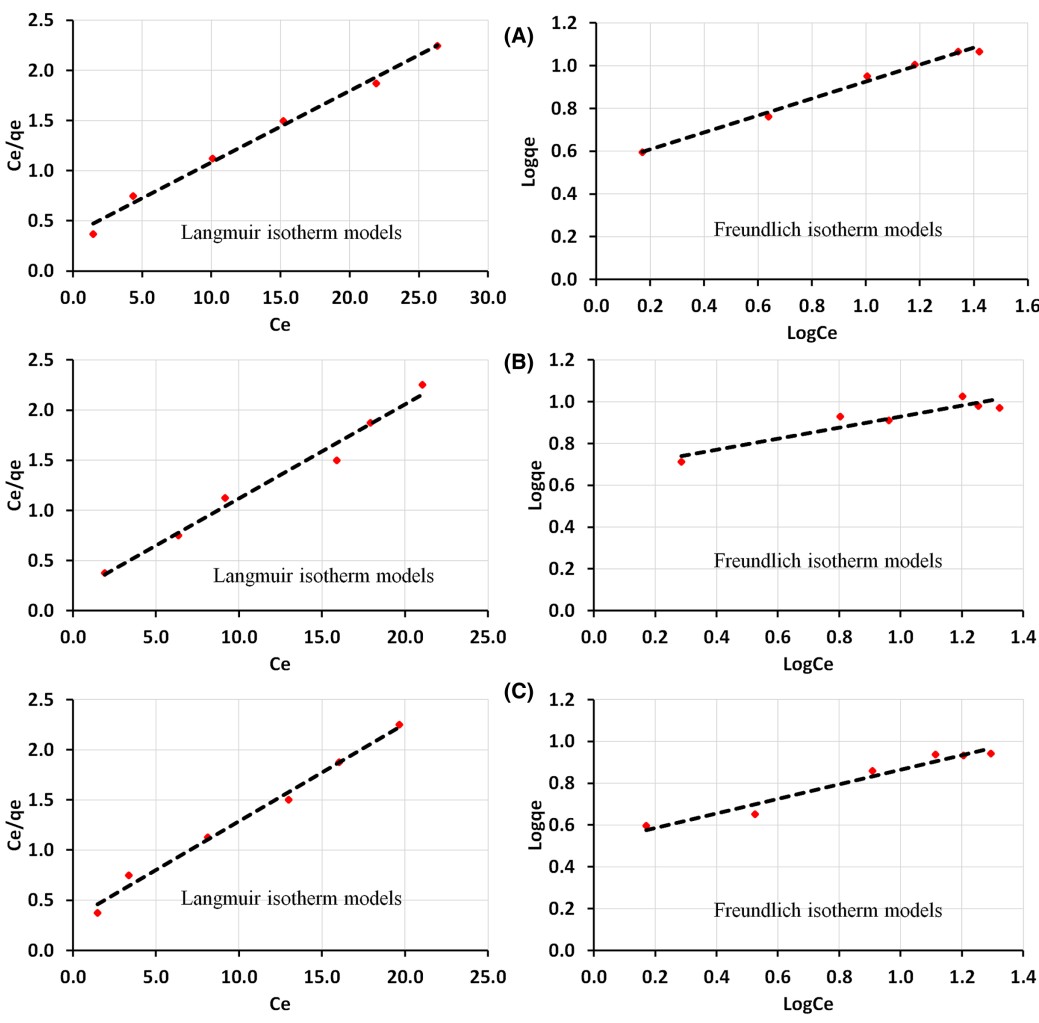

**Figure 6 Adsorption study of phenol over (A) CS–SiO₂/CaCO₃ nanocomposite (B) RHSiO₂ (C) ESCaCO₃.** Showing both Langmuir and Freundlich isotherm models. Adsorption condition: pH 5.4, and adsorbent dose of 0.15 g at 120 rpm under 60 min contact time.

Table 2 shows the correlation coefficients ($R^2$), the Langmuir constants ($qm$, and $K_L$), the Freundlich parameters ($K_F$, $1/n$), and the separation factors ($R_L$) estimated from this study.

Some authors reported materials derived from rice husk/straw to show an adsorption capacity of 13.9 (*Mandal, Mukhopadhyay & Das, 2019*). *Sarker & Fakhruddin (2017)* reported an adsorption capacity of 3.8 and 5.8 mg/g from a similar study using rice straw. *Nadavala et al. (2009)* and *Du et al. (2020)* reported an adsorption capacity of 1.26 and 8.55 mg/g using chitosan as adsorbents. About 30–58 mg/g were also reported using calcine eggshells by *Chraibi et al. (2016)* and 191 mg/g from activated carbon derived from eggshells (*Giraldo & Moreno-Piraján, 2014*). The studies reported and presented in Table 3 show phenol adsorption capacity that is either higher or lower than the values recorded in this study. However, research shows that the adsorption capacity of an

**Table 2 Langmuir and Freundlich isotherm constants for adsorption of phenol.**

| | Langmuir constants | | | Freundlich constants | | |
|---|---|---|---|---|---|---|
| Adsorbents | $qm$ (mg/g) | $K_L$ (L/mg) | $R^2$ | $K_F$ (mg/g) | $1/n$ | $R^2$ |
| CS-SiO$_2$/CaCO$_3$ | 14.06 | 0.19 | 0.99 | 3.38 | 2.52 | 0.99 |
| RHSiO$_2$ | 10.64 | 0.52 | 0.98 | 4.60 | 3.80 | 0.88 |
| ESCaCO$_3$ | 10.31 | 0.31 | 0.99 | 3.29 | 2.87 | 0.96 |
| | Separation factor ($R_L$) at various $C_i$ | | | | | |
| CS-SiO$_2$/CaCO$_3$ | 0.51 | 0.34 | 0.26 | 0.21 | 0.17 | 0.15 |
| RHSiO$_2$ | 0.79 | 0.85 | 0.88 | 0.90 | 0.92 | 0.93 |
| ESCaCO$_3$ | 0.80 | 0.79 | 0.79 | 0.78 | 0.78 | 0.79 |

**Table 3 The adsorption capacities of different adsorbents and corresponding reaction conditions.**

| Adsorbent | $qm$ (mg/g) | Adsorbent dose (g) | Adsorbent concentrations (mg/L) | Time (min) | Temp. °C | pH | Reference |
|---|---|---|---|---|---|---|---|
| Rice husk activated carbon (RHA-300) | 4.70 | 1 | 10 | 300 | 21 | 5 | *Kermani et al. (2006)* |
| Chitosan | 1.25 | 0.01 | 30 | 1,440 | 28 | 6 | *Milhome et al. (2009)* |
| Chitin | 1.96 | | | | | | |
| Natural clay | 11.09 | 0.1 | 5 | 300 | 23 | 5 | *Djebbar et al. (2012)* |
| Activated clay | 18.36 | | | | | | |
| Chitosan | 218.54 | 30 | 200 | 1,080 | 30 | 8 | *Agarwal, Sengupta & Balomajumder (2013)* |
| Activated Eggshell | 191.87 | 0.5 | 45–800 | 48 | 25 | 5.7 | *Giraldo & Moreno-Piraján (2014)* |
| Ca-bentonite | 12.5 | 1 | 125 | 30 | | 7 | *Hariani et al. (2015)* |
| Modified clays (MMT-CTAB) | 10.48 | 5 | 100 | 120 | 40 | 4 | *Ceylan, Mustafaoglu & Malkoc (2018)* |
| Rice straw | 3.75 | 2.5 | 100 | 1,440 | | 6.8 | *Sarker & Fakhruddin (2017)* |
| Eggshell | 4.00 | 4 | 5 | 80 | 20 | 3 | *Kashi (2017)* |
| Rice husk shell (RHS) | 4.35 | 2 | 15 | 60 | 25 | 5 | *Asgharnia et al. (2019)* |
| RH-activated carbon | 3.00 | | | | | | |
| Rice husk ash | 13.98 | 2 | 5 | 180 | 35 | 9 | *Mandal, Mukhopadhyay & Das (2019)* |
| Raw (RCG) clay | 1.64 | 0.2 | 30 | 180 | 60 | 4 | *Ouallal et al. (2019)* |
| Calcine (CCG) clay | 2.93 | | | | | | |
| Mg-Zn-AL(CO$_3$)hydrate clay | 12 | 10 | 40 | 1,440 | | 7 | *Tabana et al. (2020)* |
| CS-SiO$_2$/CaCO$_3$ | 14.06 | 0.15 | 25 | 60 | 28 | 5.4 | This study |

adsorbent is a function of so many parameters used in the synthesis and adsorption studies. Parameters influencing adsorption processes include the following; the initial phenol concentration, the adsorbent's characteristics, the adsorbent dose, the particle size of the adsorbent, temperature, pH, and contact time (*Mandal, Mukhopadhyay & Das, 2019*).

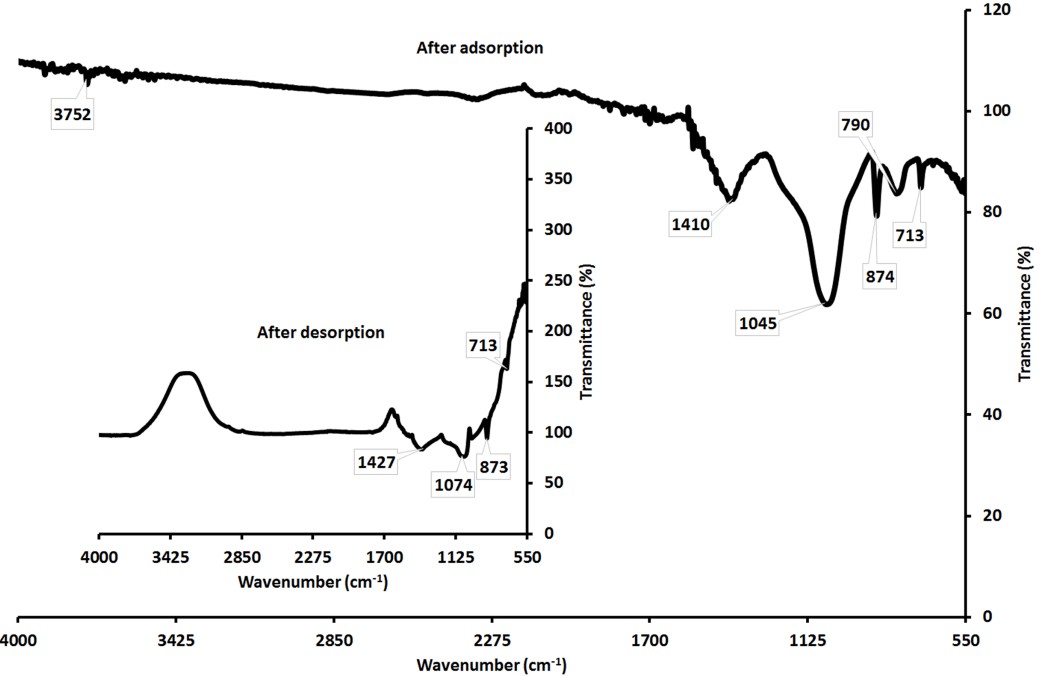

**Figure 7** Showing the FTIR of CS–SiO₂/CaCO₃ nanocomposite after adsorption of phenol and after desorption of phenol.

## Adsorption and removal mechanism of phenol over CS–SiO2/CaCO3 nanocomposite

The starting materials used in the adsorbents' synthesis (ESCaCO₃.RHSiO₂ and, CS-SiO₂/CaCO₃) are low-cost, readily available, and naturally derived from agricultural waste. Phenol adsorbed on the adsorbent can be decomposed upon re-calcination or incubation in an alkaline medium (*Tabana et al., 2020*). The removal of phenol was carried out by incubating the spent CS-SiO₂/CaCO₃ nanocomposite in 0.1 M NaOH solution for 60 min at 45 °C followed by washing using DI water (*Dehaghi et al., 2014*; *Tural, Ertaş & Tural, 2016*). Fig. 7 shows the FTIR result for CS–SiO₂/CaCO₃ nanocomposite. The study shows broadband at 3,500–3,000 cm⁻¹, and the small peak at 1,643 cm⁻¹ disappeared in CS–SiO₂/CaCO₃ nanocomposite spectra after phenol adsorption. Suggests electrostatic interaction of the phenolate with the adsorbent's hydroxide layers (*Ouallal et al., 2019*). The CS–SiO₂/CaCO₃ nanocomposite on contact with phenol further shows a change in peak intensity of $CO_3^{2-}$ at 1,410 cm⁻¹ from 1,412 cm⁻¹ before the adsorption. The phenol adsorption influences the Si–OH stretching vibration activity at 1,065 cm⁻¹ to 1,045 cm⁻¹. Similarly, a change in the asymmetric and symmetric stretching mode at 795 cm⁻¹ associated with Si–O–Si was observed on interaction with phenol to 790 cm⁻¹.

Furthermore, the spectra on the phenol's desorption from the CS–SiO₂/CaCO₃ nanocomposite show a broad contour's appearance, suggesting to be from moisture-induced crystallization (*Ouallal et al., 2019*). Moisture induced crystallization could arise from a humid environment (*Singer et al., 2012*) or at least in parts related to moisture

contents directly bound to the atoms (*Ihli et al., 2014*; *Jensen et al., 2018*) or structurally trapped in the crystals or due to fracture of hydrogen bonds formed by $H_2O...CO_3^{2-}$ hydrogen bonds. (*Cheng, Sun & Wu, 2019*; *Du & Amstad, 2020*). Certain soluble additives also facilitate moisture-induced crystallization formation (*Cheng, Sun & Wu, 2019*; *Du & Amstad, 2020*). The $CS–SiO_2/CaCO_3$ nanocomposite on phenol adsorption shows a shift in the peak at 1,410 $cm^{-1}$, 1,045 $cm^{-1}$, and 874 $cm^{-1}$, which corresponds to the peaks at 1,427 $cm^{-1}$, 1,074 $cm^{-1}$, and 873 $cm^{-1}$ after the desorption.

## CONCLUSION

A bio-based $CS–SiO_2/CaCO_3$ nanocomposite derived from low-cost waste materials (chicken eggshell, rice husk, and chitosan) were synthesized and characterized using FTIR, XRD, FESEM, and EDX. The as-synthesized materials were able to remove 84–89% of phenol in 60–90 min at a pH of 5.4, a dose of 0.15 g in 20 ml of 25 mg/L phenol concentration. The adsorption processes were observed to be complemented by both Langmuir and Freundlich adsorption processes. Suggesting good monolayer coverage on the surface of the adsorbent in addition to some heterogeneous surfaces with different available binding sites. The adsorption capacity observed in this study is 14.06 mg/g for $CS–SiO_2/CaCO_3$ nanocomposite. The FTIR analysis revealed changes in functional groups' activity before adsorption and after adsorption, suggesting good anchorage of phenol on the adsorbent. The estimation of the separation factor ($R_L$) at various initial concentrations ($Ci$) further shows the adsorption to be more favored by the Langmuir isotherm model, having values of $0 < R_L < 1$ for all the adsorbents. The research indicates that bio-based $CS–SiO_2/CaCO_3$ nanocomposite could adsorb phenol at both the monolayer and heterogeneous surfaces of $CS–SiO_2/CaCO_3$ nanocomposite.

## ACKNOWLEDGEMENTS

The authors wish to acknowledge the Department of Pure & Applied Chemistry, Faculty of Science, Adamawa State University Mubi.

### Funding

This research is supported by the Institutional Based Research (IBR) funded by Research Tertiary Education Trust Fund, Nigeria (TETFUND). The funders had no role in study design, data collection and analysis, decision to publish, or preparation of the manuscript.

### Grant Disclosures

The following grant information was disclosed by the authors:
Research Tertiary Education Trust Fund, Nigeria (TETFUND).

### Competing Interests

The authors declare that they have no competing interests.
## Author Contributions

- Ibrahim Birma Bwatanglang conceived and designed the experiments, performed the experiments, analyzed the data, performed the computation work, prepared figures and/or tables, authored or reviewed drafts of the paper, and approved the final draft.
- Samuel T. Magili analyzed the data, authored or reviewed drafts of the paper, and approved the final draft.
- Iliya Kaigamma conceived and designed the experiments, performed the experiments, analyzed the data, prepared figures and/or tables, and approved the final draft.

## Data Availability

Raw data are available in the Supplemental Files.

## Supplemental Information

Supplemental information for this article can be found online at http://dx.doi.org/10.7717/peerj-pchem.17#supplemental-information.

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
