# Peer review of "Adsorption of phenol over bio-based silica/calcium carbonate (CS-SiO₂/CaCO₃) nanocomposite synthesized from waste eggshells and rice husks"

_PeerJ Physical Chemistry, doi:10.7717/peerj-pchem.17_

## Round 0.1 · original submission · Major Revisions

On the basis of the comments from 5 independent referees, we find that your article could be of interest to the readership of Peer J. Physical Chemsitry . However, the reviewers raised concerns regarding the completeness of information provided, the adequacy of the experiments undertaken and the appropriateness of data discussion, which need proper attention. Therefore, it appears that MAJOR REVISIONS should be made before the manuscript may be considered further for publication.

We would like to invite a carefully revised manuscript that adequately addresses the concerns raised by all referees on a point-to-point basis.

You are requested to explicitly address the questions raised by reviewers and clearly point out the changes made in the manuscript. You may also include explanations for disagreeing with any of the suggestions that you have chosen not to follow. Improvement of English grammar and usage is also recommended the manuscript may be accepted.

Provide all of the above-requested information in a detailed Cover Letter when you submit the Revised Manuscript. Please also submit a Marked Manuscript in which any changes made are clearly recognizable.

We anticipate that your revised manuscript may be sent back to the original reviewers for further assessment.

Reviewer 1 ·

Basic reporting

no comment

Experimental design

no comment

Validity of the findings

no comment

Additional comments

It is not a particularly innovative work. However the work is complete and well written. So it deserves publication.

·

Basic reporting

NO comment and NO changes necessary. The article was written in conformity with any required professional standards and in accordance to the stated aim of the journal itself. Technically correct, clear and professional English was used throughout in the article. The abstract provides a concise and a complete summary of the content of the paper. Besides, the introduction provides a good, generalized background of the topic that quickly gives the reader an appreciation of the wide range of applications for this technology. Anyway, the study objectives, scientific data are clear, understandable and explicitly defined. The conclusions are mostly well supported by the results.

Experimental design

The experimental design and the scientific methods used are appropriate to the aims of the study. The article provides sufficient information for another capable researcher to reproduce the experiments described.

Validity of the findings

NO comments and NO any additional experiments are necessary to validate the results presented here. The conclusions of the study are supported by appropriate scientific-technical methods. The data on which these conclusions are based are solid, statistically sound, and controlled.

·

Basic reporting

Throughout the manuscript, the writing needs to be professional and consistent.
a. There are several incomplete or fragment sentences (e.g. lines 49, 398).

b. There are many inconsistencies with spaces (e.g. for wt.% and wt. % in lines 94 and 107, for “°C” in line 105 and 113, for “=” in lines 198 and 205, for “Fig.” in lines 166 and 192, after “CS-” in lines 402 and 409).

c. Inconsistency in subscript fonts for parameters (K1 in line 135, qm in Table 1).

d. The font of parameter symbols is not consistent with italic vs non-italic fonts. (e.g. qe in line 135 and 137).

e. Theme and bold font dissimilarities in line 138.

f. First bracket missing in equation (line 141).

g. Be consistent with theme fonts for all the equations (e.g. lines 138, 147 and more).

h. Degree symbol is missing (lines 197, 198).

i. Capital and small letters (Adsorption in line 72).

j. The period or full stop mark was twice used (lines 129).

k. Re-write the sentence in line 296-297.

l. FT-IR and FTIR (lines 120, 164, 166 and more).

Experimental design

It was mentioned that the proteins in the eggshells were denatured, washed, dried, and made into powder (line 102). The authors need to clarify the type of eggshell they used in the section of Synthesis of the Adsorbents and Characterization.

At end lines in section Adsorption Isotherms Study, the authors mentioned that the adsorption capacity of an adsorbent depends on “so many parameters” and included only three factors. The authors are suggested to provide more factors for the adsorption capacity of an adsorbent.

Validity of the findings

The authors need to provide the justification for this study in more details. Synthesis of bio-calcium carbonate from the source that this study used is not novel, as demonstrated by below mentioned papers among others. The knowledge gap being filled in this study should be clarified in Introduction section.

• Morsy, F. A.; El-Sheikh, S. M.; Barhoum, A., Nano-silica and SiO2/CaCO3 nanocomposite prepared from semi-burned rice straw ash as modified papermaking fillers. Arabian Journal of Chemistry 2019, 12 (7), 1186-1196.
• Hassan, T. A.; Rangari, V. K.; Jeelani, S., Value-added biopolymer nanocomposites from waste eggshell-based CaCO3 nanoparticles as fillers. ACS Sustainable Chemistry & Engineering 2014, 2 (4), 706-717.

Additional comments

General comments:
The authors synthesized bio-based adsorbing materials silica/calcium carbonate (CS-SiO2/CaCO3) from nature-based waste materials for removal of one of contaminant chemicals phenol from water. Though the principle behind the synthesis is not quite new, there are novelty in design and demonstration of usage for the adsorbing materials for practical phenol removal applications.

The writing needs to be more scientific and technical. There is a high chance that the reader may misinterpret or cannot understand what the authors want to explain and report. The plots in the figures have major scopes to be improved both in quality and presentation.

Reviewer 4 ·

Basic reporting

The paper is well written and interesting. The materials are not very new the synthesis is well known in the literature but the application part can be useful for the readers.

Experimental design

The experiments were well designed and executed.
It was not clear: on line 107 A solution of 5 wt. % ESCaCO3 was prepared under vigorous stirring at 3000 rpm for 30 min. What was the solvent used to prepare the %wt% solution.

It would be added advantage if authors could provide the surface area measurements of all the samples to see pore volume of the particles.

Add the CS : chitosan in the abstract

Validity of the findings

The research finding are validated with several experiments and data collected.
From the experiments its not clear that the CaCO3/ SiO2 is a physical mixture ? or any formation of calcium silicate traces. Also in XRD of composite with Chitosan the peaks related to the Chitosan are missing, pl explain the reasons.
The percentages CaCO3 and SiO2 in the mixture are advised to report
The percentage of mixture of CaCO3/ SiO2 in Chitosan should be reported.

Additional comments

The paper is well thought and experiment are well designed
It is advised to address all the comments before final publication

Reviewer 5 ·

Basic reporting

This manuscript is well organized. It was well written with clear and unambiguous, professional English used throughout.

Experimental design

Original primary research within Aims and Scope of the journal.
Research question well defined, relevant & meaningful. It is stated how research fills an identified knowledge gap.

Validity of the findings

Obtained results are original. The conclusions are appropriately stated based on investigated data.

Additional comments

1. Some specific words should be corrected or explained, such as: ESCaCO3, RHSiO2, what are ES and/or RH meaning?
2. A subsection “Chemicals and Reagents” is needed to add for indicating purity and listing of all chemicals used.
3. A subsection “Characterizations” is needed to add for listing all techniques have been used for characterization of synthesized materials.
4. All equations should be numbered and cited into body text.
5. A table must be added to compare the working conditions and the maximum adsorption capacity (Qmax) of the CS-SiO2/CaCO3 nanocomposites to other pervious reports.
6. Authors must check again the manuscript to correct all mistakes for examples: line 147, “

Annotated reviews are not available for download in order to protect the identity of reviewers who chose to remain anonymous.

---

## Round 0.2 · accepted · Accept

The revised manuscript appears to be suitable for publication as it stands.

·

Basic reporting

Pervious comments have been addressed properly. No further comment.

Experimental design

Pervious comments have been addressed properly. No further comment.

Validity of the findings

Previous comment has been addressed properly. No further comment.

Additional comments

The manuscript is better shape both in technicality and in writings. I recommend to publish the current edition without major changes.

Reviewer 5 ·

Basic reporting

Literature references, sufficient field background/context provided.
Clear and unambiguous, professional English used throughout

Experimental design

Research question well defined, relevant & meaningful. It is stated how research fills an identified knowledge gap.

Validity of the findings

All underlying data have been provided; they are robust, statistically sound, & controlled.

Additional comments

N/A